# Screening of Phytochemical, Antimicrobial, and Antioxidant Properties of *Juncus acutus* from Northeastern Morocco

**DOI:** 10.3390/life13112135

**Published:** 2023-10-29

**Authors:** Yousra Hammouti, Amine Elbouzidi, Mohamed Taibi, Reda Bellaouchi, El Hassania Loukili, Mohamed Bouhrim, Omar M. Noman, Ramzi A. Mothana, Mansour N. Ibrahim, Abdeslam Asehraou, Bouchra El Guerrouj, Mohamed Addi

**Affiliations:** 1Laboratoire d’Amélioration des Productions Agricoles, Biotechnologie et Environnement (LAPABE), Faculté des Sciences, Université Mohammed Premier, Oujda 60000, Morocco; hammouti.yousra@ump.ac.ma (Y.H.); amine.elbouzidi@ump.ac.ma (A.E.); mohamedtaibi9@hotmail.fr (M.T.); elguerroujb@gmail.com (B.E.G.); 2Centre de l’Oriental des Sciences et Technologies de l’Eau et de l’Environnement (COSTEE), Université Mohammed Premier, Oujda 60000, Morocco; e.loukili@ump.ac.ma; 3Laboratory of Bioresources, Biotechnology, Ethnopharmacology and Health, Faculty of Sciences, Mohammed First University, Boulevard Mohamed VI, B.P. 717, Oujda 60000, Morocco; r.bellaouchi@ump.ac.ma (R.B.); asehraou@yahoo.fr (A.A.); 4Laboratories TBC, Laboratory of Pharmacology, Pharmacokinetics and Clinical Pharmacy, Faculty of Pharmacy, University of Lille, 59000 Lille, France; mb@laboratoires-tbc.com; 5Laboratory of Biological Engineering, Team of Functional and Pathological Biology, Faculty of Sciences and Technology, University Sultan Moulay Slimane, Beni Mellal 23000, Morocco; 6Department of Pharmacognosy, College of Pharmacy, King Saud University, Riyadh 11451, Saudi Arabia; onoman@ksu.edu.sa (O.M.N.); rmothana@ksu.edu.sa (R.A.M.); 7Department of Agricultural Engineering, College of Food and Agriculture Sciences, King Saud University, Riyadh 11451, Saudi Arabia; malsamee@ksu.edu.sa

**Keywords:** phytochemical composition, *Juncus acutus*, HPLC-DAD, antioxidant activity, antimicrobial activity, pharmacokinetics, in silico toxicity

## Abstract

*Juncus acutus*, acknowledged through its indigenous nomenclature “samar”, is part of the Juncaceae taxonomic lineage, bearing considerable import as a botanical reservoir harboring conceivable therapeutic attributes. Its historical precedence in traditional curative methodologies for the alleviation of infections and inflammatory conditions is notable. In the purview of Eastern traditional medicine, Juncus species seeds find application for their remedial efficacy in addressing diarrhea, while the botanical fruits are subjected to infusion processes targeting the attenuation of symptoms associated with cold manifestations. The primary objective of this study was to unravel the phytochemical composition of distinct constituents within *J. acutus*, specifically leaves (JALE) and roots (JARE), originating from the indigenous expanse of the Nador region in northeastern Morocco. The extraction of plant constituents was executed utilizing an ethanol-based extraction protocol. The subsequent elucidation of chemical constituents embedded within the extracts was accomplished employing analytical techniques based on high-performance liquid chromatography (HPLC). For the purpose of in vitro antioxidant evaluation, a dual approach was adopted, encompassing the radical scavenging technique employing 2,2-diphenyl-1-picrylhydrazyl (DPPH) and the total antioxidant capacity (TAC) assay. The acquired empirical data showcase substantial radical scavenging efficacy and pronounced relative antioxidant activity. Specifically, the DPPH and TAC methods yielded values of 483.45 ± 4.07 µg/mL and 54.59 ± 2.44 µg of ascorbic acid (AA)/mL, respectively, for the leaf extracts. Correspondingly, the root extracts demonstrated values of 297.03 ± 43.3 µg/mL and 65.615 ± 0.54 µg of AA/mL for the DPPH and TAC methods. In the realm of antimicrobial evaluation, the assessment of effects was undertaken through the agar well diffusion technique. The minimum inhibitory concentration, minimum bactericidal concentration, and minimum fungicidal concentration were determined for each extract. The inhibitory influence of the ethanol extracts was observed across bacterial strains including *Staphylococcus aureus*, *Micrococcus luteus*, and *Pseudomonas aeruginosa*, with the notable exception of *Escherichia coli*. However, fungal strains such as *Candida glabrata* and *Rhodotorula glutinis* exhibited comparatively lower resistance, whereas *Aspergillus niger* and *Penicillium digitatum* exhibited heightened resistance, evincing negligible antifungal activity. An anticipatory computational assessment of pharmacokinetic parameters was conducted, complemented by the application of the Pro-tox II web tool to delineate the potential toxicity profile of compounds intrinsic to the studied extracts. The culmination of these endeavors underpins the conceivable prospects of the investigated extracts as promising candidates for oral medicinal applications.

## 1. Introduction

Infections remain a pressing global concern, posing a significant threat to public health by causing a substantial number of premature fatalities and impacting almost 50,000 individuals daily [1]. The World Health Organization underscores those illnesses stemming from microorganisms such as fungi and bacteria as a leading cause of global illness and mortality. Additionally, despite significant advancements in medical science advancements in recent times, a marked increase has been observed in the resistance of microorganisms to antibiotics [2,3,4,5,6,7]. According to a British report, bacterial resistance to antibiotics was responsible for almost 700,000 deaths worldwide in 2014 [8]. This trend of microbial resistance is escalating, with concerning forecasts from WHO projecting a potential toll of up to 10 million deaths by 2050 [7]. Nonetheless, the challenge posed by bacterial infections and the increasing resistance to specific antibiotics presents a progressively intricate therapeutic dilemma. In response to these challenges, the exploration of novel bioactive compounds from alternative sources, including medicinal plants, has become an imperative within the pharmaceutical sector [9,10,11,12]. The World Health Organization underscores that medicinal plants cater to around 80% of the health requirements of the global population [13]. These natural remedies are acknowledged for their effectiveness and the body’s favorable response to them. Plants offer a promising array of chemical variations that hold potential for uncovering novel pharmaceuticals [14], with particular emphasis on phenolic compounds. These compounds assume significance due to their diverse pharmacological attributes, including properties like cardioprotection, anticancer potential, anti-inflammatory effects, allergy alleviation, antiviral activity, vasodilation, antimicrobial action, and safeguarding against oxidative harm [15,16,17]. In light of this context, the pharmaceutical industry has shifted its focus toward exploring fresh bioactive substances derived from plants, aimed at discovering new drugs, investigating novel compounds, and refining existing ones [18,19,20].

As a result, herbal therapies have emerged as a fundamental cornerstone of primary healthcare in numerous nations. These therapies harness the curative attributes of plants, endowing them with significant value for rural communities in Africa, owing to their efficacy and economical nature [21,22,23]. Morocco, in specific, distinguishes itself as a Mediterranean nation endowed with an extensive tradition of medicinal practices and profound proficiency in employing medicinal plants. This repute is substantiated by the fact that the country is acknowledged as among the Mediterranean basin’s richest in floral and faunal diversity, as documented by Scherrer et al. in 2005 [24]. Due to its strategic geographical location and the interplay of geographical and climatic factors, the nation benefits from propitious ecological circumstances that foster the proliferation of a diverse and abundant flora. Consequently, Morocco holds a unique position for the exploration of the advantageous attributes of medicinal plants, leveraging their therapeutic potential to enhance the health and well-being of its populace [13]. Despite this botanical abundance, some plant species in Morocco have received limited scrutiny, despite exhibiting promising pharmacological effects. A notable example is *J. acutus*, commonly referred to as spiny rush, sharp rush, or sharp-pointed rush. *J. acutus*, a perennial member of the Juncaceae family, showcases densely clumped, sheath-like leaves with a simple and undivided structure. Its brown-hued flowers are arranged in either panicles or cymes, and its height ranges from approximately 30 to 200 cm [25].

This plant, known as “Azelaf” or “Semara”, is a halophyte that thrives in arid and semi-arid regions within the Mediterranean territories. Its noteworthy adaptability is evident in its ability to flourish in soils with high salinity levels, particularly in salt marshes and dune environments, where it is highly regarded for its erosion control capabilities [26]. Despite its potential, there is a notable scarcity in the scientific literature of detailed information about the phytochemical composition and specific antimicrobial attributes of *J. acutus* growing in northeastern Morocco, specifically in the Nador region.

In this pioneering study focusing on *J. acutus*, our primary aim was to explore the chemical composition, antimicrobial efficacy, antioxidative characteristics, and properties of extracts obtained through ethanol from both the leaves and roots of *J. acutus*.

## 2. Materials and Methods

### 2.1. Origin of the Plant and Extraction Method

This research employs components from both leaves and roots sourced from the northeastern region of Morocco, specifically the city of Nador. In the spring of 2023, the complete plant material was gathered and verified by a botanical specialist within the Biology Department of the Faculty of Science (FSO) at Université Mohammed Premier (UMP) in Oujda, Morocco. To initiate the process, the leaves and roots were initially ground using a grinder. Subsequently, 100 g and 40 g of the resulting leaf and root powder, respectively, were separately mixed with 99% ethanol. The resulting mixture underwent filtration using a vacuum pump, followed by solvent removal through evaporation using a rotary evaporator. This evaporation procedure occurred under controlled conditions involving 250 bar pressure, a temperature of 60 °C, and a rotational speed of 150 rpm. Following these steps, extracts were successfully obtained from both plant parts under investigation. Subsequently, these extracts were stored at 4 °C to facilitate subsequent testing.

### 2.2. Analysis of Phenolic Compounds (HPLC-DAD)

Phenolic compounds present within the extracts derived from both leaves and roots were meticulously characterized through the utilization of a high-performance liquid chromatography (HPLC) configuration. The instrumental assemblage encompassed an Agilent 1100 system (Agilent Technologies, Santa Clara, CA, USA) interfaced with a diode array UV detector from Bruker (Karlsruhe, Germany). The application of the chromatographic procedure involved the introduction of 10 μL of the extracts, subjecting them to separation within a Zorbax XDB-C18 column (5 μm, 250 × 4.6 mm) within the Agilent 1100 system. A C18 cartridge precolumn (Agilent Technologies) measuring 4 × 3 mm was placed upstream of the main column. The elution scheme was executed employing a gradient of solvents A and B: from 0 to 20 min, the composition was 80% A and 20% B, shifting to 74% A and 26% B between 20 and 56 min; subsequently, it was 24% A and 76% B between 56 and 86 min, followed by a transition to 100% B between 86 and 93 min, and concluding with 100% A between 93 and 99 min. Solvent A was an amalgamation of water and methanol (9:1) infused with 0.1% phosphoric acid, whereas solvent B was composed of methanol supplemented with 0.1% phosphoric acid. A constant flow rate of 1 mL/min was maintained for the elution process, while the system temperature was rigorously controlled at 40 °C. To discern the constituents, spectrophotometric analysis was conducted at specific wavelengths of 280 nm, enabling a comprehensive evaluation of their distinctive attributes. The identification process was substantiated by juxtaposing the retention times and UV spectra of the compounds against established reference standards, thereby facilitating their precise determination [27,28].

### 2.3. PASS and Pharmacokinetic Parameters

The pivotal determinants governing the intricate dynamics of a substance’s progression within the organism, encompassing its absorption, distribution, metabolism, and elimination (ADME), fundamentally underpin its pharmacokinetic profile. These integral facets collectively orchestrate the mechanisms by which the substance is assimilated, disseminated throughout the organism, biotransformed, and eventually excreted [27,29]. Computational methodologies are strategically harnessed to anticipate and project the ADME attributes of substances. This intricate endeavor entails the prediction of their capacity to traverse cell membranes, engage in interactions with molecules pivotal to drug absorption and excretion, and sustain their structural integrity during metabolic processes [7,27,30].

In pursuit of this objective, a comprehensive evaluation of the physicochemical attributes of the substances was undertaken, coupled with an appraisal of their congruence with established medicinal agents. The pharmacokinetic characteristics of these entities were systematically scrutinized leveraging online ADME computational tools, most notably SwissADME (http://www.swissadme.ch/ accessed on 20 July 2023) [31], and pkCSM (http://biosig.unimelb.edu.au/pkcsm/ accessed on 20 July 2023) [32]. These sophisticated platforms facilitated the dissection and projection of pertinent pharmacokinetic traits, thereby enhancing our grasp of their potential behavior within biological systems. 

### 2.4. Prediction of the Toxicity Analysis (Pro−Tox II)

For the assessment of prospective toxicity, the Pro-Tox II web utility (https://tox-new.charite.de/protox_II/ accessed on 20 July 2023) was harnessed per established protocols [33,34,35]. Employing a multifaceted approach, this application harnesses advanced statistical methodologies to establish correlations between the chemical constitution of a substance and an extensive repository of documented toxic agents. In doing so, it proactively predicts the likelihood of the substance instigating deleterious effects or adverse repercussions within human and other biological systems. This valuable tool augments our comprehension of the substance’s inherent toxicity by providing valuable insights encompassing parameters such as LD50 values, toxicity stratification, and a comprehensive spectrum of toxicological ramifications, inclusive of hepatotoxicity, carcinogenicity, immunotoxicity, mutagenicity, and cytotoxicity [33].

### 2.5. Antioxidant Activity

#### 2.5.1. Assay for Scavenging 2,2-Diphenyl-1-picrylhydrazyl (DPPH) Radicals

The assessment of the antioxidant prowess inherent to the two extracts encompassed a comparative analysis vis-à-vis a well-established antioxidant, specifically ascorbic acid. Each concentration was subjected to meticulous triadic replication for robustness. A DPPH solution was meticulously formulated at a concentration of 0.1 mM. Varied concentrations of the extract were prepared, meticulously ranging from 0.2 to 1 mg/mL. Initiating the assay, 2.5 mL of the methanolic DPPH solution intermingled with 0.5 mL from each gradient of the extract concentration. Subsequent to a 30 min period of incubation in subdued lighting, the optical density was gauged at 517 nm, normalized against an appropriate blank specimen [36]. As a positive control, the contribution of vitamin C was duly acknowledged.

The resultant scavenging activity was quantitatively ascertained through the subsequent equation: Activity (%) = (A0 − A1/A0) × 100, wherein A0 signifies the absorbance stemming from the control reaction devoid of the sample, while (A1) designates the absorbance gleaned from the extract at diverse concentrations while accompanied by the sample [7].

#### 2.5.2. Total Antioxidant Capacity (TAC)

Total antioxidant capacity (TAC) corresponds to the inherent ability of extracts to counteract the deleterious effects of free radicals and avert oxidative stress within the biological milieu, with quantification achieved through the adoption of ascorbic acid as a standard reference. The methodology instituted for the determination of TAC in this study adhered to the phosphorus-molybdenum procedure, concordant with the methodological framework outlined by Elbouzidi et al. (2023) [27]. To establish a baseline calibration, a standard curve was meticulously fashioned employing ascorbic acid, and the ensuing results were aptly quantified in terms of vitamin C equivalence.

### 2.6. Antimicrobial Activity

#### 2.6.1. Bacterial Strains and Culturing Conditions

The principal thrust of this investigation was directed towards the systematic exploration of the intrinsic antibacterial attributes exhibited by the studied extracts. The efficacy of these extract derivatives was methodically gauged through a comparative analysis encompassing the performance vis-à-vis four discrete bacterial strains. These bacterial strains were sourced from the esteemed Microbial Biotechnology Laboratory at the University of Sciences in Oujda, Morocco. The categorized strains encompassed two distinctive classifications: Gram-positive bacterial entities, notably *Staphylococcus aureus* (ATCC 6538) and *Micrococcus luteus* (LB 14110), and Gram-negative bacterial species, explicitly *Escherichia coli* (ATCC 10536) and *Pseudomonas aeruginosa* (ATCC 15442). In conformance with experimental requisites, these bacterial strains were meticulously cultured on Luria–Bertani agar medium and subsequently subjected to an incubation regimen held at 37 °C over a 24 h duration.

Prior to commencing the examination of the extracts’ antibacterial potential, a pivotal preliminary measure was diligently executed to establish homogeneity in bacterial concentration. This was artfully achieved through a meticulous quantification of bacterial density, which was subsequently standardized to a concentration of 10^6^ cells/mL. The quantification procedure leveraged the capabilities of a UV-visible spectrophotometer operating at a defined wavelength of 620 nm, facilitating precise adjustments to the bacterial content prior to the subsequent testing of the extracts. This systematic and exacting preparatory phase was meticulously executed to engender a foundation of methodological precision and result fidelity in the ensuing antibacterial appraisals.

#### 2.6.2. Agar Well Diffusion Method

The preliminary appraisal of antimicrobial efficacy against the aforementioned bacterial strains was systematically carried out utilizing the agar-well diffusion technique, as originally delineated by Perez et al. [37]. This extensively employed approach is widely acknowledged for its robustness in probing the potential to hinder microbial proliferation. This method entails the introduction of appropriately prepared bacterial inoculum onto Petri dishes laden with Mueller–Hinton agar growth medium. Small cylindrical molds with an 8 mm diameter were meticulously positioned on the surface of pre-inoculated Petri dishes, thereby generating wells. These wells were subsequently filled with 50 μL of a plant extract, suitably diluted to a concentration of 50% dimethyl sulfoxide (DMSO). Anhydrous dimethyl sulfoxide (DMSO) functioned as an emulsifying agent, facilitating the formulation of test solutions at a concentration of 1 mg/mL [38,39].

To facilitate the diffusion of the extract, the Petri dishes were judiciously preserved at 4 °C for a duration of 2 h, followed by an incubation period maintained at 37 °C for 18 to 24 h. The extent of the zone of inhibition encircling the well perimeter was meticulously measured to discern the antimicrobial efficacy against the targeted microorganisms. It is imperative to note that these rigorous evaluations were performed in triplicate for robustness and consistency.

#### 2.6.3. Measuring the MIC and the MBC 

The appraisal of substances’ efficacy in microbial control necessitates the establishment of their minimum inhibitory concentration (MIC), a pivotal parameter in this context [7,40,41]. In the present inquiry, the individual MIC values for extracts sourced from leaves and roots were discerned utilizing a resazurin microtitration assay. To facilitate this, a culture medium adherent to Mueller–Hinton specifications (comprising 0.15% aqueous content) was adroitly chosen, and the extract reservoir solutions were meticulously formulated via dissolution in DMSO. The MIC determination spanned a spectrum of antimicrobial concentrations ranging from 8% to 0.0015%, a comprehensive scope effectively evaluated through the utilization of 96-well microplates, as well as the microdilution methodology stipulated within a specified reference [42].

Precisely standardized bacterial inoculum, duly adjusted to an optimum concentration of 10^6^ cells/mL, was meticulously introduced to each well, with the inclusion of gentamicin as a positive control. Following an incubation interval of 24 h at a controlled temperature of 37 °C, the addition of a 15 µL aliquot of a 0.015% resazurin solution transpired within each well, accompanied by a subsequent 2 h incubation period maintained at 37 °C. This period elicited the transformative shift from the initial blue-hued resazurin to the concluding pink-hued resorufin, a transition meticulously monitored as per the protocol delineated within a specified reference [43]. The inviolable principle of result fidelity was adhered to through the consistent execution of each test in triplicate.

To ascertain the minimum bactericidal concentration (MBC), a minute 3 µL aliquot was meticulously transferred from wells wherein no observable growth was witnessed to a nutrient-rich cultivation medium known as Mueller–Hinton agar (MHA) [44,45]. Subsequent to this transfer, an incubation period of 24 h at a steadfast 37 °C was upheld. Upon the conclusion of this period, the MBC was elucidated as the lowest concentration wherein discernible bacterial proliferation remained conspicuously absent. As an imperative facet of ensuring the steadfastness of results, this experiment was systematically repeated across three iterations.

### 2.7. Antifungal Activity

#### 2.7.1. Selection and Source of Fungal Strains

For the assessment of antifungal efficacy, a quartet of pure fungal strains—*Penicillium digitatum, Aspergillus niger, Rhodotorula glutinis* (ON 209167), and *Candida glabrata*—were meticulously chosen from the aforementioned laboratory.

#### 2.7.2. Inoculum Cultivation and Well Diffusion Method

The cultivation protocol for *C. glabrata* and *R. glutinis* (ON 209167) involved their growth on YPD (Yeast Extract Peptone Dextrose) medium, sustained at a controlled temperature of 25 °C over a duration of 48 h. Subsequent to this period, the cellular concentration for each strain was adeptly adjusted to 10^6^ cells/mL, equating to 0.5 McFarland turbidity standards. Conversely, *P. digitatum* and *A. niger* were nurtured on BIOKAR’s PDA (potato dextrose agar) medium, maintained at 25 °C for an interval of seven days. Following this incubation, the spore density was meticulously established at 2 × 10^6^ spores/mL, employing a hematimeter (Thoma cell) as previously delineated in Section 2.5.2.

In effecting the preliminary antifungal screening, the method of well diffusion was methodically adopted, guided by the directives articulated within the Manual on Antimicrobial Susceptibility Testing [46].

#### 2.7.3. Determination of the MIC and the MFC 

In this study, the methodology of utilizing a 96-well microplate was adopted to delineate the minimum inhibitory concentrations (MICs) for the four strains under investigation. The MIC determination process encompassed a gradient of concentrations from 8% to 0.0015%. The approach applied for the MIC assessment was closely aligned with the protocol expounded upon within Section 2.6.3. The fungal suspension, approximately containing 2 × 10^8^ spores/mL, underwent an incubation phase of 48 h at 25 °C for yeast strains and 72 h for molds. Following this incubation, each well received an addition of a 15-microliter solution of resazurin to facilitate growth monitoring, followed by an additional 2 h incubation maintained at 25 °C. The MIC was subsequently established by identifying the lowest concentration at which the resazurin solution underwent a discernible transition from blue to pink, signifying the onset of growth.

For the purpose of effecting the requisite dilutions, the extracts were judiciously amalgamated with PCB containing 0.15% agar. As a comparative benchmark, the incorporation of a positive control in the form of cycloheximide was duly incorporated. To ascertain the minimum fungicidal concentration (MFC), minute samples (3 µL) were judiciously extracted from wells wherein observable growth was conspicuously absent post-MIC assessment. These samples were subsequently transposed onto YEG medium and subjected to an incubation phase of 48 h at 25 °C for yeast strains and 72 h for molds. The MFC was judiciously defined as the concentration of the extract that effectually arrested any evident fungal growth, thereby accentuating the fungicidal attributes of the extract extending beyond mere inhibition of growth.

## 3. Results and Discussion

### 3.1. Phytochemical Analysis Using HPLC-DAD

In this study, an intricate series of phytochemical analyses were meticulously conducted to elucidate the intricate chemical composition of ethanol extracts derived from *J. acutus*. The analytical methodology employed for this endeavor was the high-performance liquid chromatography (HPLC) technique (chromatograms are shown in Figure 1). Furthermore, an all-encompassing compilation, encapsulating the critical parameters of retention times and corresponding percentage areas for the identified compounds within both the leaves and roots sections, is systematically presented in the form of Table 1. 

The results revealed a group of 17 different compounds that make up the entire botanical entity under investigation. In the leaf portion, various compounds were found, encompassing gallic acid, caffeic acid, ferulic acid, catechin, hydrated catechin, syringic acid, 4-hydroxybenzoic acid, naringin, cinnamic acid, p-coumaric acid, hydrated rutin, and rutin. Conversely, the roots segment was characterized by the presence of gallic acid, hydrated catechin, syringic acid, 4-hydroxybenzoic acid, naringin, cinnamic acid, p-coumaric acid, sinapic acid, hydrated rutin, succinic acid, quercetin 3-*O*-β-D-glucoside, rutin, and kaempferol.

Among the array of compounds scrutinized, cinnamic acid emerged as the prevailing constituent within the ethanol extract derived from the leaves segment, with a proximate presence of naringin and hydrated rutin as closely trailing components. Analogously, within the roots segment, cinnamic acid similarly constituted the preeminent entity, followed in tandem by quercetin 3-*O*-β-D-glucoside and rutin. Significantly, naringin, among this assemblage of compounds, has been intrinsically linked to an assortment of health-promoting attributes [47,48]. These encompass augmentation of immune responsiveness, facilitation of DNA repair mechanisms, and adept neutralization of free radicals. Moreover, naringin has conspicuously demonstrated substantial anti-inflammatory and antioxidant characteristics, rendering it a noteworthy agent with potential utility [47]. Particularly for individuals grappling with diabetes, naringin has exhibited advantageous hypoglycemic effects, further enhancing its therapeutic significance. Cognizant attention should also be directed towards another discerned cohort of compounds, specifically cinnamic acid, acknowledged for its multifaceted spectrum of properties [49]. Notably, these include antioxidative attributes, hepatoprotective qualities, antimicrobial efficacy, mitigation of inflammatory processes, antimalarial potential, anti-tyrosinase functionality, and anticancer propensities [49,50,51]. 

Likewise, quercetin, through meticulous analysis, has unveiled its substantial antioxidative prowess, coupled with its promising attributes in the context of diabetes treatment [52,53,54]. Significantly, quercetin exhibits a noteworthy 91% inhibition of yeast α-glucosidase, a pivotal enzyme implicated in carbohydrate metabolism. Lastly, the presence of hydrated rutin within the milieu of *J. acutus* warrants particular attention, given its diverse pharmacological implications. This compound has been delineated as manifesting a multifarious portfolio of effects, encompassing antioxidative shielding, cytoprotective potency, vasoprotective utility, anticancer capacities, neuroprotection attributes, and a fortifying role in preserving cardiac health, thereby rendering it an integral constituent of considerable therapeutic potential [28,55]. Of paramount significance is the acknowledgment that the intricate biological effects exhibited by the extracts emanate predominantly from intricate and synergistic interactions amongst a constellation of chemical constituents intrinsic to the extracts, rather than stemming from the univariate impact of isolated individual compounds. The outcomes of this investigation manifest chemical compositions that mirror those delineated by El-Shamy, [55], in the context of phenolic acids, flavonoids, and coumarins. This revelation contributes to the expanding reservoir of empirical knowledge substantiating the plausible health-enhancing attributes attributed to *J. acutus* and its derivative extracts.

### 3.2. Physiochemical and Pharmacokinetic Properties (ADME) of JALE and JARE

Appendix A (see Appendix A) presents a comprehensive overview of the in-depth physicochemical and drug-likeness analyses conducted on the 17 principal compounds isolated from both JALE and JARE. Scrutinizing these analyses unveils a discernible pattern, wherein certain compounds manifest a shared presence across both extracts, while others distinctly reside within the confines of either JALE or JARE. Noteworthy constituents such as catechin, syringic acid, and 4-Hydroxybenzoic acid emerge as ubiquitous components, underscoring their pervasive distribution throughout the plant matrix.

The analytical compass of this assessment spans key parameters encompassing hydrogen-bond donors (HBDs), hydrogen-bond acceptors (HBAs), topological polar surface area (TPSA), distribution coefficient (Log Po/w), and solubility (Log S). In addition, the pivotal filters of Lipinski’s rule of five and the Veber filter were judiciously employed to meticulously gauge the tenets of drug-likeness. The outcomes of these analyses elucidate a commonality across the entire cohort of 17 compounds in terms of high solubility (designated as “+++”). However, several compounds manifest noteworthy deviations in their physicochemical attributes. An illustrative instance is evident in catechin hydrate, a shared entity within both extracts, yet it transgresses the Veber filter in JALE, wherein the tally of hydrogen bond donors or acceptors surpasses the threshold of five. In contrast, this infringement is conspicuously absent in JARE, intimating the potential for nuanced divergences in their respective applications. Crucially, all compounds harmoniously align with the stipulations delineated by Lipinski’s rule of five and the Veber filter, with the singular exception of four compounds, namely naringin (8), rutin hydrate (12), Quercetin-3-*O*-β-glucoside (14), and rutin (15).

As an illustrative instance, the compound catechin hydrate (6) is discernible within both JALE and JARE, albeit diverging in its conformance with the Veber filter. While it surpasses the stipulated threshold of five hydrogen bond donors or acceptors in JALE, this transgression is conspicuously absent in JARE, implying plausible variations in the ramifications of its utilization. Notably, specific compounds, such as naringin and rutin hydrate, consistently transgress the bounds of Lipinski’s rule of five in both extracts, owing to attributes encompassing notable molecular weight and an elevated count of hydrogen-bond acceptors. These characteristics could potentially influence their viability as prospective candidates within the realm of drug development. In contradistinction, the exclusive residence of compounds such as quercetin and kaempferol within JARE offers intriguing insights into distinctive distribution patterns and conceivable functional roles within the root constituents of *J. acutus*.

The unveiled outcomes underscore the conceivable therapeutic utility attributed to these compounds, primarily anchored in their physicochemical attributes and harmonious alignment with the prescribed tenets of drug-likeness. Nonetheless, it remains imperative that comprehensive in vitro and in vivo inquiries be undertaken to rigorously validate both the efficacy and safety profiles of these compounds.

Appendix A (see Appendix A) provides an illustrative exposition of the ADME (absorption, distribution, metabolism, and excretion) attributes inherent to the 17 preeminent compounds distinctly present within the JALE and JARE extracts. These essential parameters have been systematically classified into discrete categories encompassing absorption parameters, distribution parameters, metabolism parameters, and excretion parameters. Notably, this comprehensive compilation accentuates the bioavailability score, a pivotal metric employed to prognosticate the proportion of an orally administered compound that successfully traverses the barriers of absorption to achieve systemic circulation. The bioavailability score, encapsulating a numeric spectrum between 0 and 1, is indicative of the degree of enhanced bioavailability, with values approximating 1 denoting heightened efficacy in this regard [56]. 

In an examination of the absorption parameters, a triad of pivotal metrics emerges, encompassing the bioavailability score, Caco-2 permeability, and intestinal absorption (%). Within this schema, the bioavailability score, spanning the spectrum from 0 to 1, stands as a reflection of the proportion of the administered dose that can traverse intact into the bloodstream. Elevated scores, as demonstrated by compounds such as ferulic acid, syringic acid, and cinnamic acid, denote a heightened potential for enhanced bioavailability, thereby rendering these entities more amenable to effective therapeutic utilization.

Caco-2 permeability, a predictive measure predicated on the Caco-2 cell model [57,58,59], augments this narrative by discerning the intestinal permeability of a given compound. Positive values manifest a propensity for favorable permeability, with notables such as caffeic acid, ferulic acid, and 4-Hydroxybenzoic acid exemplifying this trait, suggesting efficient absorption within the intestinal milieu.

Transitioning to the domain of distribution parameters, the tandem of Log Kp (cm/s) and VDss emerges as pivotal determinants, delineating the extent of compound dispersion within the anatomical confines. Evidently, most compounds are characterized by Log Kp values around −2.7, indicative of restrained distribution. Distinct nuances, however, arise, notably discernible in quercetin, which exhibits a marginally lower value, connoting a relatively circumscribed distribution scope. In contradistinction, VDss unveils a contrasting facet, divulging the volume of distribution at steady state. Catechin (4), naringin (8), rutin hydrate (12), quercetin 3-*O*-β-D-glucoside (14), rutin (15), quercetin (16), and kaempferol (17) stand as exemplars of pronounced dispersion within the body.

Analyzing BBB permeability assesses the potential of compounds to cross the blood–brain barrier, with Log BBB values exceeding 0.3 indicating successful penetration. Interestingly, none of the examined compounds fall within this range. However, compounds like catechin hydrate and naringin exhibit relatively lower values, suggesting a possible limitation in their ability to cross the blood–brain barrier.

In the realm of prospecting for imminent drug metabolism and potential toxicity, a foundational stride involves the meticulous evaluation of the projected conduct and interactions of a given molecule with cytochrome P450 (CYP) isozymes. Within this purview, the term “conduct” assumes a connotation that encompasses the multifaceted influences of a molecule upon biological systems, encapsulating its proclivity to selectively engage with designated receptors or enzymes. The interplay between a molecule and CYP isozymes, pivotal protagonists orchestrating the biotransformation of myriad pharmaceutical agents, furnishes invaluable insights into the latent pharmacokinetic attributes that the molecule may exhibit [60].

Evidently, the ensemble of identified compounds remains bereft of reactivity as substrates for the cytochrome P450 (CYP) enzymes, notably CYP2D6 and CYP3A4, distinguished for their cardinal roles in drug metabolism [60]. By the same token, the compounds under scrutiny evince an absence of inhibitory influence upon the enzymatic activities of CYP2D6 and CYP3A4.

In the intricate framework of drug disposition, the Renal Organic Cation Transporter 2 (OCT2) assumes a pivotal mantle in expediting the renal excretion of diverse pharmaceutical entities. In instances where a drug assumes the role of an OCT2 substrate, its urinary clearance can be augmented, thereby potentiating its renal elimination. This facet of efficacious drug excretion, facilitated through the instrumental agency of prominent organic cation transporters like Renal OCT2, bears undeniable significance in the landscape of drug metabolism. Nonetheless, the compounds that have come to light conspicuously lack the salient attributes indicative of their capacity to serve as substrates for OCT2 [61].

The comprehensive examination of compound clearance takes into account the intricate interplay between both hepatic and renal pathways [32], and the observed outcomes are meticulously detailed in Appendix A (see Appendix A). This tabular exposition substantively confers pivotal insights into the ADME (absorption, distribution, metabolism, and excretion) characteristics intrinsic to the identified compounds resident within the JALE and JARE extracts. Such insights, by virtue of their fundamental nature, immeasurably enrich the capacity to prognosticate the pharmacological comportment of these compounds and their potential amenability in the sphere of drug developmental endeavors.

In the endeavor to gauge the foreseeable oral bioavailability of the identified phytoconstituents, an analytical lens was cast upon six distinctive physicochemical attributes, each manifested through the medium of bioavailability radar plots. These attributes, encompassing facets such as lipophilicity, dimensions, polarity, solubility, flexibility, and saturation, collectively wield pivotal influence in delineating the potential of a molecule to be assimilated and effectively harnessed within the intricate milieu of biological systems. Appendix A (in the Appendix A) serves as an illustrative pictorial medium that illuminates the bioavailability radar profiles intrinsic to the identified compounds, thereby encapsulating a tangible graphical rendition of their plausible oral bioavailability.

The demarcation of the discernible pink region within the radar plot assumes significance as it embodies the criteria requisite for a molecule to achieve drug-like status, an imperative determinant in the assessment of the molecule’s viability as a prospective therapeutic agent. This delineation distinctly signifies the propensity for efficient absorption and widespread dispersion within the biological milieu. The strategic employment of bioavailability radar plots confers invaluable insights into the prospective oral bioavailability of compounds, thereby facilitating the identification of promising candidates meriting subsequent phases of drug development and therapeutic deployment.

Within the specific purview of this investigation, which centers upon the discerned phytoconstituents, the entirety of their physicochemical attributes align harmoniously with the designated pink radar region within the bioavailability radar, with the exception of the presence of insaturation noted within molecules 1, 2, 3, 4, 5, 6, 7, 9, 10, 11, 16, and 17. This observation alludes to the potential prevalence of double bonds or unsaturated functional groups within the structural composition of these compounds. Furthermore, compounds 8, 12, 14, and 15 manifest deviations from the stipulated polarity criterion, implying a potential heightened polarity that might impede effective translocation through lipid membranes within the gastrointestinal tract. Consequently, this polarity variance could potentially result in a diminished scope for bioavailability attainment.

The BOILED-Egg model (Appendix A, see Appendix A) presents an initial framework for the preliminary assessment of a molecule’s proclivity for intestinal absorption and penetration across the blood–brain barrier. This model hinges on two fundamental parameters: lipophilicity, as quantified by WLOGP, and polarity, measured by TPSA [62]. The graphical representation of this model illustrates a white sector denoting molecules with heightened potential for intestinal absorption, while the yellow yolk region signifies molecules with a higher likelihood of traversing the blood–brain barrier [62]. Within this visualization, distinct coloration is employed to differentiate molecules that act as substrates for P-glycoprotein from those that do not. Blue dots symbolize substrate molecules, while red dots represent non-substrates. In the specific context at hand, a select quartet of phytocompounds (namely ferulic acid (3), 4-Hydroxybenzoic acid (7), cinnamic acid (9), and p-coumaric acid (10)) have emerged as displaying noteworthy potential for absorption and proficient passage across the blood–brain barrier. Furthermore, the analysis underscores that all of these compounds, with the exception of catechin (4) (as illustrated in Appendix A, see Appendix A), do not function as substrates for P-glycoprotein.

### 3.3. In Silico Toxicity Prediction (Using Pro-Tox II)

Appendix A (see Appendix A) presents the toxicological evaluation of prominent compounds identified in both JALE and JARE extracts. The enumerated compounds include gallic acid, caffeic acid, ferulic acid, catechin, syringic acid, catechin hydrate, 4-Hydroxybenzoic acid, naringin, cinnamic acid, *p*-coumaric acid, sinapic acid, rutin hydrate, succinic acid, quercetin 3-*O*-β-D-glucoside, rutin, quercetin, and kaempferol. Foreseen LD50 values (expressed in mg/kg) and the categorization according to GHS hazard classes are furnished, accompanied by their respective probabilities for diverse toxic endpoints [33].

The LD50 values provide crucial insights into the gradient of toxicity, revealing that compounds 1, 3, 5, 8, 10, 11, 12, 13, and 14 exhibit a propensity towards low acute toxicity, in contrast to compounds 4 and 9, which demonstrate higher acute toxicity. Within the framework of GHS hazard classifications ranging from III to VI, the majority of compounds are situated within hazard classes IV and V. These outcomes suggest that while most compounds within the JALE and JARE extracts exhibit low toxicity, certain constituents may pose potential hazards to human health.

The anticipated toxicological endpoints unveil potential health implications associated with exposure to these compounds, including hepatotoxicity, mutagenicity, carcinogenicity, immunotoxicity, and cytotoxicity. Notably, compounds 4 and 6 exhibit an enhanced likelihood of hepatotoxicity and mutagenicity. Compound 6 also emerges as a probable candidate for heightened carcinogenicity, while compounds 4 and 14 demonstrate moderate probabilities of carcinogenic potential. Immunotoxicity is predicted for compounds 4, 6, and 7, with compound 6 displaying a particularly high likelihood. Moreover, cytotoxicity is envisaged across various compounds, with compounds 1, 5, 6, and 7 displaying the most pronounced probabilities.

These findings provide pertinent insights into potential health implications associated with key compounds present in the JALE and JARE extracts, thus significantly contributing to the judicious and safe utilization of these agents. However, the imperative for further research and investigations remains inevitable to establish a comprehensive understanding of their toxicity profiles and corresponding implications for human health.

The toxicological assessment of primary compounds inherent to JALE and JARE compositions reveals a diverse spectrum of toxicity levels and hazard classes. While certain entities exhibit low toxicity profiles and are classified as non-toxic, others manifest higher toxicity magnitudes, potentially carrying risks associated with ingestion. The distinct toxicological endpoints, encompassing hepatotoxicity, carcinogenicity, immunotoxicity, mutagenicity, and cytotoxicity, serve as pivotal indicators, shedding light on the landscape of secure usage and application of JALE and JARE extracts. The imperative need for further exploratory endeavors and comprehensive investigations emerges as a guiding principle to achieve a comprehensive understanding of their toxicity characteristics and the consequent impact on human health. This scientific knowledge stands poised to maximize the manifold benefits offered by these extracts while concurrently minimizing potential health contingencies.

### 3.4. Antioxidant Activity

In this study, an exhaustive assessment of the antioxidant capacity of *J. acutus* was systematically executed by employing the 2,2-diphenyl-1-picrylhydrazyl (DPPH) free radical scavenging technique and the total antioxidant capacity (TAC) assay. The computation of the IC50 value, representing the quantity of antioxidants required to achieve a 50% reduction in free radical levels, was diligently carried out using both methodologies. The results distinctly revealed pronounced antioxidant efficacy within the roots extract, as evidenced by IC50 values of 297.03 ± 43.3 µg/mL for the DPPH assay and 65.615 ± 0.54 µg ascorbic acid/mL for the TAC assay (refer to Table 2 for detailed data). Concurrently, the leaves of *J. acutus* exhibited significant proficiency in quenching the stable DPPH radical to DPPH-H, exemplified by IC50 values of 483.45 ± 4.07 µg/mL for the DPPH assay and 54.59 µg ascorbic acid/mL ± 2.44 for the TAC assay.

The observed outcomes indicate that the roots exhibit a notably prominent total antioxidant capacity (TAC) and display substantial proficiency in scavenging DPPH radicals. These activities are ascribable to the presence of polyphenolic constituents, including sinapic acid and cinnamic acid, along with flavonoids such as rutin, naringin, and quercetin-3-*O*-β-glucoside. Disparities in antioxidant efficacy between the two extracts can likely be attributed to their distinct compositional profiles. Significantly, these findings align coherently with prior investigations that scrutinized the antioxidant potential of *J. acutus* leaves and roots through methanolic extraction in the context of the DPPH assay. These earlier studies substantiated the existence of bioactive compounds within the methanolic extracts of both *J. acutus* leaves and roots, capable of counteracting free radicals [63].

### 3.5. Antibacterial Activity

In the course of this investigation, we executed a series of experiments encompassing two distinct extract types: one derived from the roots and the other sourced from the leaves of the *J. acutus* plant. The fundamental objective of these experiments was to assess and characterize the antimicrobial attributes inherent to these extracts. To accomplish this, the well diffusion assay was deployed as a pivotal tool, facilitating the quantification of zones of inhibition surrounding the wells. These zones of inhibition serve as tangible indicators of the extracts’ capacity to impede the growth of bacteria in their immediate vicinity.

For a more comprehensive understanding of the antimicrobial efficacy, we additionally employed the microdilution technique. This method facilitated the precise determination of both the minimum inhibitory concentrations (MICs) and the minimum bactericidal concentrations (MBCs) associated with the extracts. The MIC corresponds to the lowest concentration of an extract that effectively hampers bacterial growth, while the MBC signifies the minimal concentration at which bacterial proliferation is not only inhibited but also culminates in bacterial cell death. This systematic approach allows us to precisely gauge the potency of these extracts against the targeted microorganisms.

The outcomes of our investigation unveil noteworthy inhibitory properties exhibited by both the extracts derived from the roots and leaves of *J. acutus* against select bacterial strains, namely *S. aureus* and *M. luteus* (Table 3). Of particular significance is the leaves extract’s remarkable efficacy in inhibiting Staphylococcus aureus and *P. aeruginosa*, as manifested by inhibition zones measuring 21 mm and 15 mm, respectively. In contrast, the roots extract showcases heightened potency in suppressing *M. luteus* growth, yielding an inhibition zone measuring 20 mm. Notably, neither extract demonstrates inhibitory effects on the bacterial strain *E. coli*.

The outcomes of our study closely mirror the observations elucidated by Gachkar et al. (2007), wherein they elucidated the variations in inhibitory efficacy against Gram-positive and Gram-negative bacteria due to the distinctive attributes of their cell wall compositions. In the context of Gram-positive bacteria, the inhibitory mechanism predominantly hinges upon the direct interaction of hydrophobic compounds with phospholipids present within the cell membrane [64,65]. This interaction may potentially result in structural perturbations, complete membrane disintegration, and the disruption of internal cellular components. Conversely, the inherent resistance exhibited by Gram-negative bacteria arises from diffusion limitations across their outer membranes, giving rise to a hydrophilic barrier that impedes the entry of inhibitory agents [66].

Notably, both extracts exhibited substantial inhibition of *S. aureus* growth even at a minimal concentration of 2% (*v/v*). In contrast, their effectiveness against the *P. aeruginosa* strain was relatively attenuated, necessitating higher concentrations with minimum inhibitory concentrations reaching 8%. The leaves extract demonstrated inhibitory activity at a concentration of 2% against *M. luteus*, while the roots extract required a concentration of 4% to achieve inhibition.

The most remarkable discovery pertains to the extracts’ notable efficacy against *S. aureus*, underscoring their potential as potent antimicrobial agents against this bacterial strain. At concentrations of 4%, both extracts exhibited bactericidal activity against *S. aureus*. Particularly, the leaves extract displayed enhanced potency, achieving inhibition against *M. luteus* and *P. aeruginosa* strains at concentrations of 8% and 16%, respectively. In contrast, the roots extract exhibited relatively diminished efficacy, necessitating concentrations of 16% and higher for the inhibition of *M. luteus* and *P. aeruginosa* strains.

The underlying rationale for these findings is closely intertwined with the chemical composition of *J. acutus* extracts. The HPLC-DAD profiling of the plant highlighted the presence of hydroxycinnamic acids and flavonoids as the predominant constituents. Hydroxycinnamic acids, encompassing cinnamic acid, caffeic acid, ferulic acid, and sinapic acid, are renowned for their antimicrobial attributes against diverse microorganisms. Similarly, flavonoids are acknowledged for their potent antimicrobial properties and their role in safeguarding plants against pathogenic agents [28].

Flavonoids operate through multifaceted mechanisms, encompassing the inhibition of cell envelope synthesis, modulation of efflux pumps, interference with nucleic acids, attenuation of virulence enzymes, disruption of biofilm formation, and perturbation of microbial membranes. Thus, the prevalence of hydroxycinnamic acids and flavonoids within *J. acutus* extracts offers a plausible rationale for the observed favorable outcomes in terms of antimicrobial activity [67].

### 3.6. Antifungal Activity

The botanical extract exhibited noteworthy antifungal efficacy against *C. glabrata* and *R. glutinis* fungal strains, evinced by inhibition zone diameters spanning 11 to 12 mm (refer to Table 4). Remarkably, exclusive antifungal inhibition against *C. glabrata* was observed solely in the roots extract, manifesting a substantial 12 mm inhibition zone. This notable effect can be ascribed to the substantial presence of quercetin 3-*O*-β-D-glucoside within the roots extract, attaining a remarkable concentration of 22.48%. This specific phytochemical entity is distinguished for its potent biological activities, particularly its well-established antibacterial and antifungal attributes, which evidently underpin the robust performance of the roots extract [28]. Of particular interest, the minimum inhibitory concentration (MIC) against both fungal strains were consistently determined at 4%, while the minimum fungicidal concentration (MFC) surpassed 16% for both extracts. This pattern suggests that the extracts possess antifungal efficacy against yeast species while concurrently refraining from eliciting inhibitory effects on mold species. This divergent response may potentially be rooted in the emergence of resistant spores within sporulating molds, thereby conferring an adaptive capacity for survival in adverse environmental conditions [68]. Furthermore, this nuanced variance in antifungal activity could potentially stem from the intricate interplay of multiple factors, synergistically contributing to the attainment of effective antifungal efficacy.

It is pivotal to underscore that the conspicuous lack of inhibitory activity against molds might feasibly be attributed to their sporulating nature, intricate metabolic responses, and the underlying structural attributes of their cellular constitution [69].

## 4. Conclusions and Future Perspectives

In the realm of antimicrobial exploration, a comprehensive investigation was embarked upon, directed towards two distinct extracts derived from *J. acutus*. The outcomes of this diligent endeavor unveiled a potent inhibitory prowess intrinsic to these extracts. Evidently, their pronounced inhibitory effects were more conspicuous against bacterial strains in comparison to fungal strains, with a notable proclivity for Gram-positive bacteria. The meticulous analysis of the chemical constitution enveloping the extracts obtained from the *J. acutus* plant ushered forth a revelation of paramount significance. Within this botanical treasure trove, a cadre of hydroxycinnamic acids and flavonoids emerged as key players, those including cinnamic acid, quercetin 3-*O*-β-D-glucoside, kaempferol, and syringic acid. These compounds, steeped in renowned repute, harbor an expansive spectrum of biological attributes, spanning domains such as antidiabetic, antibacterial, and antifungal actions. Moreover, the discerning eye of this study unveiled a noteworthy facet: variations in both the chemical composition and biological efficacy contingent upon the specific plant part harnessed.

The abundant variety of natural metabolites found in the ethanol extracts of *J. acutus* presents an enticing opportunity as medicinal alternatives, particularly showcasing antioxidant and antibacterial properties. Looking ahead, a promising and thoughtful approach lies in the field of bio-guided research. This approach involves exploring the intricate chemical composition of the plant to identify and isolate key components with significant pharmacological potential, such as the renowned cinnamic acid. These substances not only hold the potential to provide preventive and therapeutic solutions for various ailments but also offer intriguing possibilities in areas like nutraceuticals and food products. The significant pharmacological importance of these compounds enhances their potential to act as transformative agents, driving innovation and enhancing the landscape of health and well-being. In terms of future perspectives, this research opens up exciting opportunities. Further exploration of *J. acutus* and its metabolites can lead to the development of novel medicines and health-promoting products. Additionally, ongoing investigations can help uncover new applications for these compounds, potentially revolutionizing various sectors and contributing to advancements in healthcare and well-being.

## Figures and Tables

**Figure 1 life-13-02135-f001:**
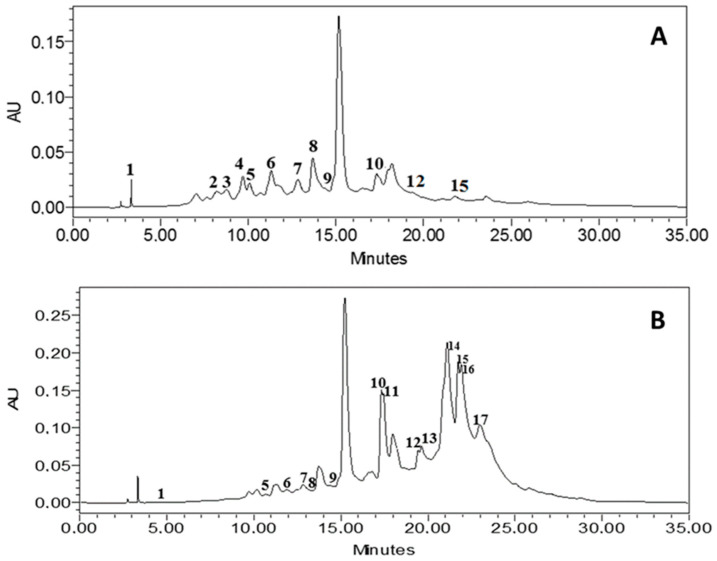
HPLC-DAD chromatograms of the phenolic composition of JALE (**A**) and JARE (**B**). Note: the corresponding molecules are in Table 1.

**Table 1 life-13-02135-t001:** Phenolic profile of the ethanolic extract of leaves and roots from *J. acutus* using HPLC-DAD. n.d. Not determined.

N°	Compounds	Formula	Group	RT (min)	%Area	
					Leaves	Roots
**1**	Gallic acid	C_7_H_6_O_5_	Phenolic acids	4.912	2.03	1.73
**2**	Caffeic acid	C_9_H_8_O_4_	Hydroxycinnamic acids	8.216	3.76	n.d.
**3**	Ferulic acid	C_10_H_10_O_4_	Hydroxycinnamic acids	8.767	n.d.	3.86
**4**	Catechin	C_15_H_14_O_6_	Flavonoids (Flavan-3-ols)	9.700	n.d.	4.66
**5**	Catechin hydrate	C_15_H_14_O_6_•H_2_O	Flavonoids (Flavan-3-ols)	10.622	1.35	2.64
**6**	Syringic acid	C_9_H_10_O_5_	Hydroxybenzoic acids	11.685	1.47	3.87
**7**	4-Hydroxybenzoic acid	C_7_H_6_O_3_	Hydroxybenzoic acids	12.839	0.64	4.87
**8**	Naringin	C_27_H_32_O_14_	Flavonoids (Flavanones)	13.516	3.28	12.16
**9**	Cinnamic acid	C_9_H_8_O_2_	Hydroxycinnamic acids	14.433	24.1	44.71
**10**	*p*-coumaric acid	C_9_H_8_O_3_	Hydroxycinnamic acids	17.321	8.20	4.77
**11**	Sinapic acid	C_11_H_12_O_5_	Hydroxycinnamic acids	17.483	4.20	n.d
**12**	Rutin hydrate	C_27_H_30_O_16_•xH_2_O	Flavonoid glycosides	19.113	6.86	11.93
**13**	Succinic acid	C_4_H_6_O_4_	Dicarboxylic acids	19.432	3.16	n.d.
**14**	Quercetin 3-*O*-β-D-glucoside	C_21_H_20_O_12_	Flavonoid glycosides	21.104	22.48	n.d.
**15**	Rutin	C_27_H_30_O_16_	Flavonoid glycosides	21.912	6.93	0.86
**16**	Quercetin	C_15_H_10_O_7_	Flavonoids	21.912	13.25	n.d.
**17**	Kaempferol	C_15_H_10_O_6_	Flavonoids	22.964	3.07	n.d.

**Table 2 life-13-02135-t002:** Assessment of the antioxidant potential of JALE and JARE by DPPH and TAC assays, values are expressed as mean ± SEM (n = 3).

Antioxidant Test	Inhibitory Concentration 50 (IC_50_)		
	JALE	JARE	Ascorbic Acid (AA)
DPPH assay (µg/mL)	483.45 ± 4.07	297.03 ± 43.3	44.37 ± 3.56
TAC (µg AA/mL)	54.59 ± 2.44	65.615 ± 0.54	-

**Table 3 life-13-02135-t003:** Antibacterial activity results for the two ethanolic extracts of *J. acutus*.

Bacterial Strain		*S. aureus*	*M. luteus*	*P. aeruginosa*
Diameter of inhibition zone (mm)	JALE	21 ± 1	18 ± 0.4	15 ± 0.5
	JARE	19 ± 0.5	20 ± 0.5	13 ± 1
Gentamicine (1 mg/mL)		24	25	28
MIC (% *v/v*)	JALE	2	2	8
	JARE	2	4	8
MBC (% *v/v*)	JALE	4	8	16
	JARE	4	16	>16

All values in this table represent the mean ± SD (n = 3).

**Table 4 life-13-02135-t004:** Antifungal activities of the two ethanolic extracts of *J. acutus.* All values in this table represent the mean ± SD (n = 3).

Bacterial Strain		*C. glabrata*	*R. glutinis*
Diameter of inhibition zone (mm)	JALE	0 ± 0	11.2 ± 0.4
	JARE	12 ± 0.5	11.4 ± 0.1
Cycloheximide, IZ (1 mg/mL)		24 ± 0.5	21.0
MIC (% *v/v*)	JALE	-	4
	JARE	4	4
MFC (% *v/v*)	JALE	-	>16
	JARE	>16	>16

## Data Availability

Not applicable.

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
