# Peer review of "Screening of Phytochemical, Antimicrobial, and Antioxidant Properties of *Juncus acutus* from Northeastern Morocco"

_life, 2023, doi:10.3390/life13112135_

Round 1
Reviewer 1 Report
Comments and Suggestions for Authors
The work done by author is good. need some changes before final publishing
1- in introduction give upto date references and compare your work with others.
2- show novality
3- in conclusion give future aspects also
4- take care while describing units, see SI System of units.
5- references need format
Comments on the Quality of English Language
see grammar mistakes, as some where sentences are much long and some show lack of actual meaning
Author Response
Dear editors and reviewers,
We would like to express our gratitude for giving us the opportunity to improve our manuscript through the revised version, and we sincerely appreciate your valuable comments. We are particularly grateful to the reviewers for their comprehensive review, which helped improve the manuscript's quality.
In response to the requested changes, we have carefully addressed each query and weakness by incorporating the suggested changes or providing a detailed response. Major changes to the revised manuscript have been highlighted in yellow for ease of identification. We assure you that all linguistic concerns and typos have been rectified in the manuscript, although they may not be mentioned explicitly in this response.
Once again, we sincerely appreciate your time, effort and constructive feedback. We hope our revised manuscript successfully addresses all of the reviewers' comments and meets the necessary criteria for publication in “Life” journal.
Thank you, kind regards
Reviewer 1.
The work done by author is good. need some changes before final publishing
We express our gratitude to reviewer 1 for her/his valuable comments and recommendations, which have significantly enhanced the quality of our manuscript, making it eligible for publication in the Life journal.
- In introduction give up to date references and compare your work with others.
Answer: Thank you for your insightful feedback. We have incorporated your suggestions into the introduction section of our work. We have added a paragraph that highlights the therapeutic effects of the plant, drawing upon research conducted by various recent researchers.
- Show novelty.
Answer: In accordance with your recommendations, significant modifications have been made to our work. A new section has been inserted in the introduction to highlight the uniqueness of our study, emphasizing that it will be the first research conducted on this plant in the Oriental region.
- In conclusion give future aspects also
Answer: In response to the reviewer's request to include future aspects in the conclusion, we have added passages highlighted in yellow to highlight perspectives to be explored. Thank your for your pertinent suggestion.
- take care while describing units, see SI System of units.
Answer: We have rectified this issue. All the units used in our study are in accordance to the International System of Units. Thank you
- References need format
Answer: we have added the references according to the journal’s guidelines.

Reviewer 2 Report
Comments and Suggestions for Authors
I have a problem with this manuscript. It is well-written and complex, yet most of the exploited computational techniques are not scientifically sound in my opinion. For instance, it is very difficult to properly assess the hemotoxicity of a given compound without taking into account the formulation process, protonation forms, self-association potency, etc. The calculation of these parameters is of course possible, but it rarely corresponds to the real-world applicability of the compunds under study. The way I see it, Tables 2 (wrong caption), 3 and 4, as well as Figure 2 should be moved to Supplementary Information, whereas mostly the experimental data should remain in the main manuscript.
Comments on the Quality of English LanguageThe language is perhaps too 'shakespearean' - very nice and poetic, yet in some sections it is uneasy to extract the main thought behind the written text.
Author Response
Dear editors and reviewers,
We would like to express our gratitude for giving us the opportunity to improve our manuscript through the revised version, and we sincerely appreciate your valuable comments. We are particularly grateful to the reviewers for their comprehensive review, which helped improve the manuscript's quality.
In response to the requested changes, we have carefully addressed each query and weakness by incorporating the suggested changes or providing a detailed response. Major changes to the revised manuscript have been highlighted in yellow for ease of identification. We assure you that all linguistic concerns and typos have been rectified in the manuscript, although they may not be mentioned explicitly in this response.
Once again, we sincerely appreciate your time, effort and constructive feedback. We hope our revised manuscript successfully addresses all of the reviewers' comments and meets the necessary criteria for publication in “Life” journal.
Thank you, kind regards
Reviewer 2.
We express our gratitude to reviewer 2 for her/his valuable comments and recommendations, which have significantly enhanced the quality of our manuscript, making it eligible for publication in the Life journal.
- I have a problem with this manuscript. It is well-written and complex, yet most of the exploited computational techniques are not scientifically sound in my opinion. For instance, it is very difficult to properly assess the hemotoxicity of a given compound without taking into account the formulation process, protonation forms, self-association potency, etc. The calculation of these parameters is of course possible, but it rarely corresponds to the real-world applicability of the compunds under study. The way I see it, Tables 2 (wrong caption), 3 and 4, as well as Figure 2 should be moved to Supplementary Information, whereas mostly the experimental data should remain in the main manuscript.
Answer: We would like to express our gratitude for your thoughtful feedback regarding our manuscript. Your comments and concerns are highly appreciated. It's essential to clarify that our study did not specifically target the assessment of hemotoxicity. Instead, our primary focus revolved around evaluating organ toxicity, with a particular emphasis on hepatotoxicity, and examining critical toxicological aspects such as immunotoxicity, mutagenicity, and carcinogenicity. We acknowledge your perspective on the computational techniques utilized in our study. It's worth noting that the computational methods we employed are predominantly rooted in statistical models derived from experimental data. For instance, in the case of hepatotoxicity assessment, we adopted the Pro-Tox II model, which was trained using a dataset of over 850 compounds for the training set and more than 90 compounds for the test set. The method employed in this model is Random Forest with SMOTE TC (Synthetic Minority Over-Sampling using Tanimoto Coefficient) sampling, for more details on the adopted computational techniques in our study please refer to the materials and methods section in our manuscript.
Once again, we thank you for your valuable comments and suggestions. We will consider your proposal to relocate certain tables and figures to the Supplementary Information section to enhance the clarity and focus of our main manuscript. Your input is highly significant, and we have diligently worked on addressing these aspects to enhance the overall quality of our research.
- The language is perhaps too 'shakespearean' - very nice and poetic, yet in some sections it is uneasy to extract the main thought behind the written text.
Answer: We have revised the language in our manuscript, aiming to make intricate sections more accessible and easier to understand. Thank you for your comment.

Reviewer 3 Report
Comments and Suggestions for Authors
This study aims to investigate the phytochemical composition, antimicrobial effectiveness, antioxidative traits, and properties of the extracts obtained through ethanol from both the leaves and roots of Juncus acutus from Northeastern Morocco, specifically the Nador region.
The Materials and Methods are very complete, and the Results and Discussion are well described, presented with three figures and seven tables, and the most important findings are well connected and discussed with references to other studies.
The results revealed that the root and leaf of J. acutus could be promising leads for the discovery of new therapeutic agents.
There are some minor changes that need to be made but which are of the presentation nature. The proposed review changes are as follows:
- On page 10, line 416, “Table 1” should be “Table 2”;
- From page 11 onwards there is a pagination numbering error;
- On line 652, “Table 2” should be “Table 6”;
- On lines 677 and 678, “Micrococcus luteus” and “Pseudomonas aeruginosa” should be in italic;
- On line 695, “Candida glabrata” should be in italic;
- On line 714, “Table 3” should be “Table 7”.
With these minor changes, the recommendation will be to accept the manuscript for publication.
Author Response
Dear editors and reviewers,
We would like to express our gratitude for giving us the opportunity to improve our manuscript through the revised version, and we sincerely appreciate your valuable comments. We are particularly grateful to the reviewers for their comprehensive review, which helped improve the manuscript's quality.
In response to the requested changes, we have carefully addressed each query and weakness by incorporating the suggested changes or providing a detailed response. Major changes to the revised manuscript have been highlighted in yellow for ease of identification. We assure you that all linguistic concerns and typos have been rectified in the manuscript, although they may not be mentioned explicitly in this response.
Once again, we sincerely appreciate your time, effort and constructive feedback. We hope our revised manuscript successfully addresses all of the reviewers' comments and meets the necessary criteria for publication in “Life” journal.
Thank you, kind regards
Reviewer 3.
This study aims to investigate the phytochemical composition, antimicrobial effectiveness, antioxidative traits, and properties of the extracts obtained through ethanol from both the leaves and roots of Juncus acutus from Northeastern Morocco, specifically the Nador region.
The Materials and Methods are very complete, and the Results and Discussion are well described, presented with three figures and seven tables, and the most important findings are well connected and discussed with references to other studies. The results revealed that the root and leaf of J. acutus could be promising leads for the discovery of new therapeutic agents.
Response: We would like to express our gratitude for your thoughtful feedback regarding our manuscript. Your comments and concerns are highly appreciated.
There are some minor changes that need to be made but which are of the presentation nature. The proposed review changes are as follows:
- On page 10, line 416, “Table 1” should be “Table 2”; On line 652, “Table 2” should be “Table 6”; On line 714, “Table 3” should be “Table 7”.
Response: We have rectified tables numbering throughout the manuscript. Note that some tables were moved to the supplementary file. Thank you for your sharp remark.
- From page 11 onwards there is a pagination numbering error;
Response: We have rectified pagination numbering. Thank you for your remark.
- On lines 677 and 678, “Micrococcus luteus” and “Pseudomonas aeruginosa” should be in italic; On line 695, “Candida glabrata” should be in italic;
Response: All scientific names were written in Italics, thank you for your remark.
With these minor changes, the recommendation will be to accept the manuscript for publication.
Response: We sincerely appreciate your insightful comments and feedback on our manuscript. Your thoughtful analysis of our study is both encouraging and valuable, and helped us improve the quality of our work.
